# Characteristics of MDR *E. coli* strains isolated from Pet Dogs with clinic diarrhea: A pool of antibiotic resistance genes and virulence-associated genes

Yu Yuan[1☯], Yan Hu[1☯], Xiaoli Zhang[2☯], Wenhao Zhong[1☯], Shulei Pan[1], Liqin Wang[3], Ziyao Zhou[1], Haifeng Liu[1], Shaqiu Zhang[1], Guangneng Peng[1], Ya Wang[1], Qigui Yan[1], Yan Luo[1], Keyun Shi[2]*, Zhijun Zhong[1]*

1 College of Veterinary Medicine, Sichuan Agricultural University, Key Laboratory of Animal Disease and Human Health of Sichuan, Chengdu, China, 2 Jiangsu Yixing People's Hospital, Yixing, China, 3 The Chengdu Zoo, Institute of Wild Animals, Chengdu, China

☯ These authors contributed equally to this work.
* zhongzhijun488@126.com (ZZ); staff955@yxph.com (KS)

**Data Availability Statement:** All relevant data are within the paper and its Supporting Information files.

## Abstract

The increasing number of multi-drug resistant (MDR) bacteria in companion animals poses a threat to both pet treatment and public health. To investigate the characteristics of MDR *Escherichia coli* (*E. coli*) from dogs, we detected the antimicrobial resistance (AMR) of 135 *E. coli* isolates from diarrheal pet dogs by disc diffusion method (K-B method), and screened antibiotic resistance genes (ARGs), virulence-associated genes (VAGs), and population structure (phylogenetic groups and MLST) by polymerase chain reaction (PCR) for 74 MDR strains, then further analyzed the association between AMRs and ARGs or VAGs. Our results showed that 135 isolates exhibited high resistance to AMP (71.11%, 96/135), TET (62.22%, 84/135), and SXT (59.26%, 80/135). Additionally, 54.81% (74/135) of the isolates were identified as MDR *E. coli*. In 74 MDR strains, a total of 12 ARGs in 6 categories and 14 VAGs in 4 categories were observed, of which *tetA* (95.95%, 71/74) and *fimC* (100%, 74/74) were the most prevalent. Further analysis of associations between ARGs and AMRs or VAGs in MDR strains revealed 23 significant positive associated pairs were observed between ARGs and AMRs, while only 5 associated pairs were observed between ARGs and VAGs (3 positive associated pairs and 2 negative associated pairs). Results of population structure analysis showed that B2 and D groups were the prevalent phylogroups (90.54%, 67/74), and 74 MDR strains belonged to 42 STs (6 clonal complexes and 23 singletons), of which ST10 was the dominant lineage. Our findings indicated that MDR *E. coli* from pet dogs carry a high diversity of ARGs and VAGs, and were mostly belong to B2/D groups and ST10. Measures should be taken to prevent the transmission of MDR *E. coli* between companion animals and humans, as the fecal shedding of MDR *E. coli* from pet dogs may pose a threat to humans.

**Funding:** This research was funded by the National Key Research and Development Program of China (2018YFD0500900, 2016YFD0501009), the Chengdu Giant Panda Breeding Research Foundation (CPF2017-05, CPF2015-4) and the Science and Technology Achievements Transfer Project in Sichuan province (2022JDZH0026). The funders had no role in study design, data collection and analysis, decision to publish, or preparation of the manuscript.

**Competing interests:** The authors have declared that no competing interests exist.

## 1. Introduction

*Escherichia coli* (*E. coli*) is one of commensal microbiota in the gut of humans and animals [1]. With the widespread use of antimicrobials, the occurrence of multi-drug resistance (MDR) *E. coli* in humans and animals has posed a major threat to public health [2,3]. The presence of MDR *E. coli* in companion animals, such as pet dogs, undoubtedly raises concerns for both dogs and humans due to the high antimicrobial resistance (AMR) and the capability of carrying various antibiotic resistance genes (ARGs) of MDR strains [4,5]. Furthermore, previous studies have shown that there were associations between ARGs and virulence-associated genes (VAGs) existed in *E. coli*, and ARG-carrying strains may increase the likelihood of carrying VAGs [6]. However, only limited studies focused on MDR strains from pets in China, especially in Sichuan province which is considered as a major province for keeping pets in Southwest China [7].

In addition, recent studies have found that *E. coli* clones can be extensively shared between humans and household animals [8,9]. The transmission of high-risk *E. coli* clones between animals and humans has been recognized as a major public health issue [10,11]. For the population structure of *E. coli* clones, phylogenetic studies have shown that the B2 and D groups were more prevalent than the commensal groups (A and B1 groups) in *E. coli* isolates from diarrheic pet dogs [12,13]. Moreover, the multilocus sequence typing (MLST) was also used to analyze the population structure of *E. coli* clones [14]. Recent studies have identified a high population diversity of STs which related to zoonotic or pathotype in *E. coli* isolates from pet dogs, such as ST354, ST393, and ST457 *E. coli* were observed in companion animals from Australia [15]. And the high-risk ExPEC clones associated with humans and multidrug resistance, including sequence type (ST) 38, ST131, ST224, ST167, ST354, ST410, ST617 and ST648, have been identified in cats and dogs in Thailand which suggested there may be clonal dissemination between pets and humans [16].

In 2023, the number of domestic pets has exceeded 100 million in China [7]. Moreover, the latest annual report shows that the use of veterinary antibiotics for animals in China has exceeded 30,000 tons in 2020, tetracyclines, β-lactamases, and sulfonamides antimicrobial agents were widely used (Veterinary Bulletin of the Ministry of Agriculture and Rural People's Republic of China, 2020) [17].With the widespread use of antimicrobials in animals, the high prevalence of MDR *E. coli* isolates from pet dogs has raised health concerns for both companion animals and humans [18,19]. To better understand the characteristics of MDR *E. coli* from pet dogs, we analyzed the antimicrobial resistance of *E. coli* isolates from diarrheal dogs, focusing on ARGs, VAGs, and population structure (ST and phylogenetic groups) for MDR *E. coli* to evaluate the potential threat.

## 2. Materials and methods

### 2.1 Sample collection

A total of 185 fresh feces samples were collected from pet dogs with clinical diarrhea during August 2021 to June 2022. Samples were excluded if the pet dogs were prescribed antimicrobial therapy or veterinary admission within the previous 3 months. The study was permitted by committee of Sichuan agricultural university (Permission number: DYY-2020303164) and the Sichuan Agricultural University Animal Ethical and Welfare Committee (Permission number: 20210268).

All samples were collected by professional veterinarians in Veterinary Teaching Hospital of Sichuan Agricultural University. Disinfection of hands and changing of disposable gloves were mandatory before sample collection. Samples were taken from the center of fresh feces by using sterile cotton swabs as soon as the pet dogs defecated to avoid cross-contamination from

environmental bacteria on the ground. Then, samples were collected in sterile micro centrifuge tubes (Eppendorf) and were sent for processing to the laboratory within 12 h after collection.

## 2.2 Strains isolation

The isolation and identification of *E. coli* was performed as previous studies described [20,21]. Fecal samples were enriched in Luria-Bertani (LB) broth at 37˚C, 120 r/min in a shaking incubator for 24 h. All isolates were identified presumptively by using phenotypic methods, including Gram staining, MacConkey agar growth and Eosin Methylene Blue agar growth. We further used 16S rDNA sequences (Primer: 5'-GAGTTTGATCCTGGCTCAG-3'; 5'-AGAAAGG AGGTGATCCAGCC-3') [22] for final identification of *E. coli*. The confirmed isolates were stored in Luria-Bertani (LB) broth containing 50% glycerol at −20˚C for further analysis.

## 2.3 Screening of MDR strains by antimicrobial susceptibility test

The antimicrobial susceptibilities of all isolates were tested using the standard disk diffusion method recommended by the Clinical and Laboratory Standards Institute (CLSI). A total of 16 antimicrobial agents in 6 categories as below were tested. aminoglycosides (gentamicin, CN, 10 μg; tobramycin, TOB, 10 μg), tetracyclines (tetracycline, TET, 30 μg; doxycycline, DOX, 30 μg), amide alcohols (chloramphenicol, C, 30 μg), quinolones (ciprofloxacin, CIP, 5 μg), sulfonamides (trimethoprim-sulfamethoxazole, SXT, 10 μg), β-lactams (ampicillin, AMP, 10 μg; cefazolin, KZ, 30 μg; cefuroxime, CXM, 30 μg; cefotaxime, CTX, 30 μg; cefepime, FEP, 30 μg; cefoxitin, FOX, 30 μg; aztreonam, ATM, 30 μg; imipenem, IPM, 10 μg; amoxicillin/clavulanic acid 2:1, AMC, 20 μg). For the 16 antimicrobial agents, the CN and TOB in aminoglycosides, DOX in tetracyclines, KZ, CTX, AMP, AMC in β-lactams and CIP in quinolones were used in pet animals at the location of our present study. The other antimicrobial agents (TET, C, CXM, FEP, ATM, IPM, FOX and SXT) have been found with *E. coli* resistance in dogs according to previous studies [23–25]. Results were interpreted in accordance with CLSI criteria (CLSI, 2023) [26]. *E. coli* ATCC25922 was used as a control. MDR strain was defined as being resistant to three or more antimicrobial categories [20].

## 2.4 DNA extraction and detection of ARGs, VAGs

Total genomic DNA of strains was extracted from isolates by using TIANamp Bacteria DNA kit (Tiangen Biotech, Beijing, China) following the manufacturer's instructions. DNA samples were stored at -20˚C for subsequent polymerase chain reaction (PCR) detection.

Primers of 23 ARGs in 6 categories (including 5 $bla_{CTX-M}$ alleles for group1, 2, 8, 9, and 25) and 25 VAGs in 5 categories were synthesized by Huada Gene Technology Co. Ltd (Shenzhen, China). The PCR primer sequences and conditions for the ARGs and VAGs were showed in S1 and S2 Tables, respectively. PCR products were separated by gel electrophoresis in a 1.0% agarose gel in 1 × TAE buffer (40 mM Tris-acetate, 1 mM EDTA, pH 8.3), stained with Gold-View^TM (Sangon Biotech, Shanghai, China), and photographed under ultraviolet light using the Bio-Rad ChemiDoc MP omnipotent imager (Bole, USA). All positive PCR products were sequenced with Sanger sequencing in both directions by Sangon Biotech (Shanghai, China). Sequences of ARGs and VAGs were analyzed online using the BLAST function of NCBI (http://blast.ncbi.nlm.nih.gov).

## 2.5 Phylogenetic grouping and MLST

Phylogenetic grouping (A/B1/B2/D) was determined for MDR *E. coli* isolates using PCR targeting *chuA*, *yjaA*, and *TSPE4.C2* following the protocol of Clermont et al. [27]. MLST was

based on the sequencing results of seven housekeeping genes (*adk*, *fumC*, *gyrB*, *icd*, *mdh*, *purA* and *recA*) in each strain to obtain a numerical allelic profile which is abbreviated to a unique identifier of each strain: sequence type (ST) [28]. Allelic types of all seven housekeeping genes and ST of strains were determined following the protocol of the *E. coli* MLST database (https://pubmlst.org) [28,29]. The primer sequences of genes for MLST and phylogroups were shown in S3 Table. PCR products were separated by gel electrophoresis in a 1.0% agarose gel stained with GoldViewTM and photographed under ultraviolet light. All positive PCR products were sequenced with Sanger sequencing in both directions by Sangon Biotech (Shanghai, China). The sequences of housekeeping gene for MLST were analyzed online using pubMLST database (https://pubmlst.org).

The goeBURST algorithm in phyloviz 2.0 was used to clustering analysis of STs for MDR *E. coli* isolates, which divided the STs into several clusters, named as clonal complexes (CC), which consist of closely related STs with two allelic differences [30]. A clonal complex is typically composed of a single predominant genotype and closely relatives genotype [31].

## 2.6 Association analysis between ARGs and Antimicrobial resistance phenotypes (AMRs) or VAGs

The statistically analysis of data of AMR, ARGs and VAGs was conducted by using SPSS Statistics (version 26.0). *P*-value < 0.05 was considered to be statistically significant. Association analysis between ARGs and AMR or VAGs was performed by ggplot2 in RStudio (version 4.2.2; http://www.r-project.org) [32].

## 3. Results

### 3.1 AMRs of 135 isolates and Screening of MDR strains

A total of 135 *E. coli* strains were successfully isolated from 185 fecal samples of diarrheal dogs. Among the 135 *E. coli* isolates, 118 strains (87.41%, 118/135) were resistant to at least one antimicrobial agent, while only 17 (12.59%, 17/135) strains were sensitive to all antimicrobial agents (Fig 1A). Among the 6 antibiotic categories, the resistance rate for β-lactam antibiotics had the highest resistance rats (76.30%, 103/135), followed by tetracyclines (64.44%, 87/135) and sulfonamides (59.26%, 80/135). The resistance rate to quinolones antibiotics was the

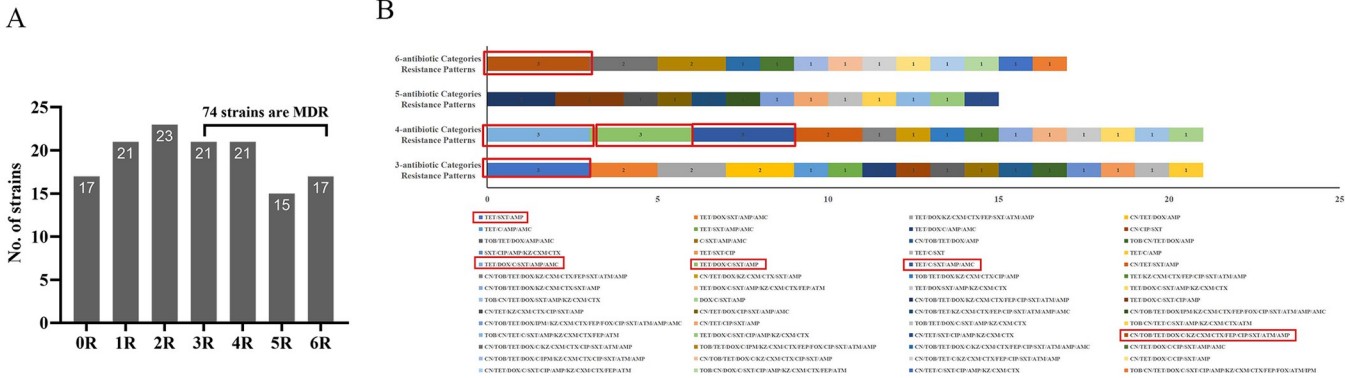

**Fig 1. Antibiotic resistance patterns of *E. coli* isolates from pet dogs.** (A) The abscissa 0R represents the strains that were sensitive to all antibiotics, and 1–6 R represents strains that were resistant to 1–6 antibiotic categories, respectively. Seventy-four *E. coli* isolates are MDR, of which 17 strains were resistant to 6 antibiotic categories; (B) Color bars demonstrate the distribution of phenotypic resistance patterns in 74 MDR *E. coli* isolates, and the Arabic number represent the number of strains. A total of 56 resistance patterns were observed by using disc diffusion assay. The red boxes highlight the prevalent resistant-phenotypes patterns (occurring three times), the other combinations (without red box) occurred only once or twice.

**Table 1. Antimicrobial resistance (AMR) detected in *E. coli* strains isolated from pet dogs (n = 135).**

| Category of antimicrobial | No. of Resistant Isolates (%) | Antibiotic | | No. of Resistant Isolates (%) |
|---|---|---|---|---|
| β-lactams | 103(76.30) | Penicillins | Ampicillin (AMP) | 96 (71.11) |
| | | 1rd/2nd generation Cephalosporins | Cefazolin (KZ) | 48 (35.56) |
| | | | Cefuroxime (CXM) | 44 (32.59) |
| | | 3rd/4th generation Cephalosporins | Cefotaxime (CTX) | 45 (33.33) |
| | | | Cefepime (FEP) | 21 (15.56) |
| | | β-lactam compound | Amoxicillin/clavulanic acid (AMC) | 32 (23.70) |
| | | Monobactams | Aztreonam (ATM) | 25 (18.52) |
| | | Carbapenems | Imipenem (IPM) | 6 (4.44) |
| | | Cephamicins | Cefoxitin (FOX) | 5 (3.70) |
| Tetracyclines | 87(64.44) | Tetracycline (TET) | | 84 (62.22) |
| | | Doxycycline (DOX) | | 61 (45.19) |
| Sulfonamides | 80(59.26) | Trimethoprim-sulfamethoxazole (SXT) | | 80 (59.26) |
| Aminoglycosides | 45(33.33) | Gentamicin (CN) | | 39 (28.89) |
| | | Tobramycin (TOB) | | 28 (20.74) |
| Amide alcohols | 41(30.37) | Chloramphenicol (C) | | 41 (30.37) |
| Quinolones | 35(25.93) | Ciprofloxacin (CIP) | | 35 (25.93) |

lowest (25.93%, 35/135). Moreover, the resistance rates to AMP (71.11%, 96/135), TET (62.22%, 84/135), and SXT (59.62%, 80/135) were the top 3 in 16 antimicrobial agents. The lowest resistance rate was observed for FOX (3.70%, 5/135), and the resistance rates for remaining 12 antibiotics ranged from 4.44% (IPM) to 45.19% (DOX) (Table 1). We further analyzed the resistant-phenotype patterns of the 135 *E. coli* isolates, revealing that 54.81% (74/135) of isolates were identified as MDR *E. coli* (Fig 1A). Among the 74 MDR strains, 56 types of resistance phenotypic patterns were observed. The more common patterns were TET/SXT/AMP, TET/DOX/C/SXT/AMP/AMC, TET/DOX/C/SXT/AMP, TET/C/SXT/AMP/AMC, and CN/TOB/TET/DOX/C/KZ/CXM/CTX/FEP/CIP/SXT/ATM/AMP (Fig 1B).

## 3.2 Distribution of ARGs and VAGs in MDR *E. coli* strains

Twelve out of 18 ARGs in 6 categories were detected in our present study (Table 2 and Fig 2). The detection rate of *tetA* (95.95%, 71/74) was the highest, followed by $bla_{TEM}$ (93.24%, 69/74) and $bla_{CTX-M}$ (90.54%, 67/74). Moreover, detection rates of *flor*, *qnrS*, and *sul2* were all over 70% (74.32%, 74.32% and 70.27%, respectively). The detection rates of the remaining ARGs ranged from 16.22% (*oqxAB*) to 44.59% (*sul1*).

We further analyzed the subtypes of $bla_{TEM}$ and $bla_{CTX-M}$, 3 variants of the $bla_{TEM}$ gene and 5 variants of the $bla_{CTX-M}$ gene were detected. Among the 3 variants of the $bla_{TEM}$ gene, $bla_{TEM-1}$ was the most frequent (81.16%, 56/69), followed by $bla_{TEM-135}$ (15.94%, 11/69) and $bla_{TEM-176}$ (2.90%, 2/69). For the $bla_{CTX-M}$ gene, 3 variants in $bla_{CTX-M-1}$ group ($bla_{CTX-M-55}$, $bla_{CTX-M-15}$ and $bla_{CTX-M-64}$) and 2 variants in $bla_{CTX-M-9}$ group ($bla_{CTX-M-65}$ and $bla_{CTX-M-14}$) were detected, $bla_{CTX-M-14}$ (79.10%, 53/67) and $bla_{CTX-M-55}$ (31.34%, 21/67) were more prevalent among the 5 variants observed, followed by $bla_{CTX-M-64}$ (7.46%, 5/67), $bla_{CTX-M-15}$ (5.97%, 4/67) and $bla_{CTX-M-65}$ (4.45%, 3/67).

Fourteen out of 25 VAGs in 4 categories were detected in our study (Table 3 and Fig 2). The detection rate of *fimC* (100%, 74/74) was the highest, followed by *iucD* (90.54%, 67/74) and *sitA* (85.14%, 63/74). The detection rates of *iss* (72.97%, 54/74), *ompT* (71.62%, 53/74) and *fyuA* (71.62%, 53/74) were all above 70%. Detection rates of the remaining VAGs were ranged

**Table 2. Details of antimicrobial resistance (AMR) and antibiotic resistance genes (ARGs) detected in MDR *E. coli* strains isolated from pet dogs (n = 74).**

| Category of antimicrobial | No. of Resistant Isolates (%) | Antibiotics | No. of Resistant Isolates (%) | ARGs | No. of Positive Isolates (%) | Alleles of $bla_{TEM}$/$bla_{CTX-M}$ (%) |
|---|---|---|---|---|---|---|
| β-lactams | 71 (95.95) | AMP | 71 (95.95) | $bla_{TEM}$ | 69(93.24) | $bla_{TEM-1}$, 81.16 (56/69) |
| | | KZ | 38 (51.35) | | | $bla_{TEM-135}$, 15.94 (11/69) |
| | | CTX | 38 (51.35) | | | $bla_{TEM-176}$, 2.90 (2/69) |
| | | CXM | 38 (51.35) | $bla_{CTX-M}$ | 67(90.54) | $bla_{CTX-M-14}$, 79.10 (53/67) |
| | | ATM | 25 (33.78) | | | |
| | | FEP | 21 (28.38) | | | $bla_{CTX-M-65}$, 4.45 (3/67) |
| | | AMC | 19 (25.67) | | | $bla_{CTX-M-55}$, 31.34 (21/67) |
| | | IPM | 6 (8.11) | | | $bla_{CTX-M-64}$, 7.46 (5/67) |
| | | FOX | 5 (6.76) | | | $bla_{CTX-M-15}$, 5.97 (4/67) |
| Tetracyclines | 71 (95.95) | TET | 69 (93.24) | *tetA* | 71(95.95) | – |
| | | DOX | 49 (66.22) | | | – |
| Sulfonamides | 64 (90.54) | SXT | 64 (90.54) | *sul2* | 52(70.27) | – |
| | | | | *sul1* | 33(44.59) | – |
| | | | | *sul3* | 13(17.57) | – |
| Aminoglycosides | 2 (56.76) | CN | 37 (50.00) | *aacC2* | 52(70.27) | – |
| | | TOB | 28 (37.84) | *aacC4* | 31(41.89) | – |
| Amide alcohols | 40 (54.05) | C | 40 (54.05) | *flor* | 55(74.32) | – |
| | | | | *cmlA* | 19(25.68) | – |
| Quinolones | 34 (45.95) | CIP | 34 (45.95) | *qnrS* | 55(74.32) | – |
| | | | | *oqxAB* | 12(16.22) | – |

from 4.05% (*aggR*) to 54.05% (*eae*). Further analysis of the virulence determinants linked with different pathotypes revealed 3 DEC-related VAGs (*eae*, *aggR*, *astA*) and 11 ExPEC-related VAGs were detected. Notably, all MDR *E. coli* isolates carried at least one ExPEC-related VAG, 75.68% (56/74) of strains carried at least one DEC-related VAG, which also carried at least one ExPEC-related VAG. The average number of VAGs carried by per DEC-related strain (8.36) was significantly higher than that of DEC-unrelated VAGs strains (5.56) ($P < 0.01$).

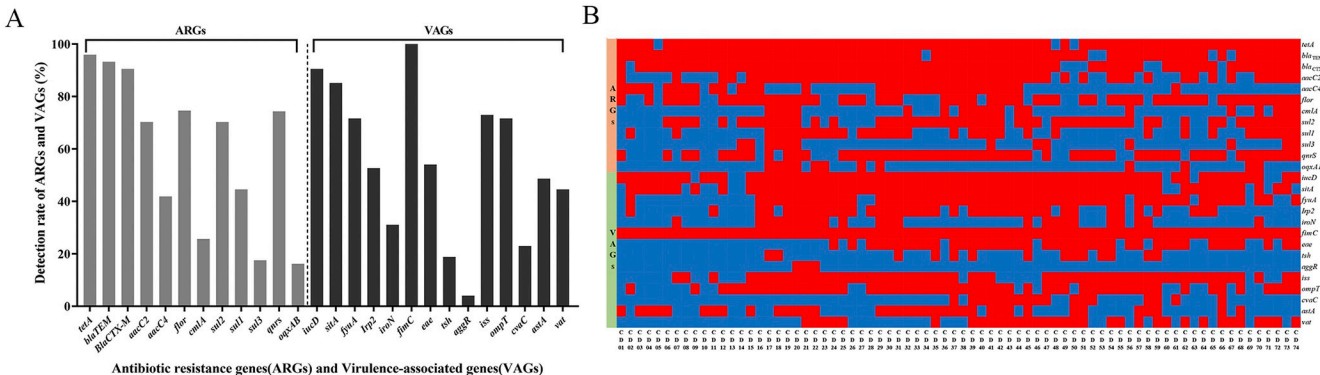

**Fig 2. Distribution of antibiotic resistance genes (ARGs) and virulence-associated genes (VAGs) in 74 MDR *E. coli* strains from pet dogs.** (A) The bar graphs show the detection rates of ARGs and VAGs. A total of 12 ARGs and 14 VAGs were detected, of which *tetA* (95.95%) and *fimC* (100%) were the most prevalent; (B) The abscissa represents the ID of MDR isolates and ordinate represents ARGs and VAGs. The red and blue regions represent the presence or absence of corresponding ordinate genes in an isolate, respectively. A high diversity of ARGs and VAGs was detected among MDR strains.

**Table 3. Detection rate of virulence-associated genes (VAGs) in MDR *E. coli* isolated from pet dogs (n = 74).**

| Category of virulence | VAGs | No. of Resistant Isolates (%) | Related pathotypes |
|---|---|---|---|
| Iron transport-related | *iucD* | 67 (90.54) | ExPEC |
| | *sitA* | 63 (85.14) | ExPEC |
| | *fyuA* | 53 (71.62) | ExPEC |
| | *Irp2* | 39 (52.70) | ExPEC |
| | *iroN* | 23 (31.08) | ExPEC |
| Adhesion-related | *fimC* | 74 (100.00) | ExPEC |
| | *eae* | 40 (54.05) | DEC |
| | *tsh* | 14 (18.91) | ExPEC |
| Invasion-and-toxin related | *vat* | 33 (44.59) | ExPEC |
| | *astA* | 36 (48.65) | DEC |
| | *aggR* | 3 (4.05) | DEC |
| Antiserum survival factors-related | *iss* | 54 (72.97) | ExPEC |
| | *ompT* | 53 (71.62) | ExPEC |
| | *cvaC* | 17 (22.97) | ExPEC |

## 3.3 Associations between ARGs and AMRs or VAGs in MDR *E. coli* strains

Details of the detection rates of ARGs and AMR in 74 MDR *E. coli* strains were showed in Table 2. High prevalence of the β-lactam antibiotics genes (*bla*$_{TEM}$ and *bla*$_{CTX-M}$, 93.24% and 90.54%, respectively) were detected, and resistance rate of β-lactams antibiotic (AMP) was higher than 90% (71/74, 95.95%). For tetracyclines, *tetA* (71/74, 95.95%) was detected in MDR strains, and the proportions of strains with tetracyclines-resistant phenotypes were 95.95%. For sulfonamides, the resistant rate of SXT was 90.54% (64/74) in MDR strains, while detection rates of three related resistance genes (*sul1*, *sul2* and *sul3*) ranged from 17.57% (13/74) to 70.27% (52/74). For aminoglycosides, only two resistance genes (*aacC2* and *aacC4*) were detected with the detection rates ranging from 41.89% (31/74) to 70.27% (52/74), while the detection rate of MDR strains with aminoglycosides-resistant phenotype (CN and TOB) ranged from 37.87% (28/74) to 50% (37/74). Similarly, the detection rates of remaining antibiotic resistance phenotypes and related ARGs were fluctuated. Above results showed that only β-lactams and tetracyclines antibiotic-resistant phenotypes were generally matched with related ARGs in detection rates, but the others were not completely consistent.

We further analyzed the associations between ARGs and AMR in 74 MDR strains. As shown in Fig 3A, a total of 23 positive association pairs ($r > 0$, $P < 0.05$) were observed between ARGs and AMR, of which the association between amide alcohols antibiotic resistance gene *flor* and amide alcohols-resistant phenotype (C) was the strongest. Moreover, the associations between 12 ARGs and 14 VAGs were further analyzed and the results were showed in Fig 3B, only 5 association pairs were observed, of which 3 pairs (*sul2/ompT*, *aacC2/ompT*, *aacC4/astA*) showed positive association ($r > 0$, $P < 0.05$), and the strongest association pair was observed between *sul2* and *ompT*. The other 2 pairs (*bla*$_{CTX-M}$/*tsh*, *bla*$_{CTX-M}$/*cvaC*) were negative associations ($r < 0$, $P < 0.05$), and the strongest negative association pair was observed between *bla*$_{CTX-M}$ and *tsh*.

## 3.4 Phylogenetic Grouping and MLST of 74 MDR *E. coli* isolates

As shown in the Fig 4A, group B2 (71.62%, 53/74) was the most prevalent in 74 MDR *E. coli* strains, followed by group D (18.92%, 14/74) and group B1 (9.46%, 7/74). 90.54% (67/74) of the MDR *E. coli* strains belonged to virulent extraintestinal-related group (B2 and D) and only

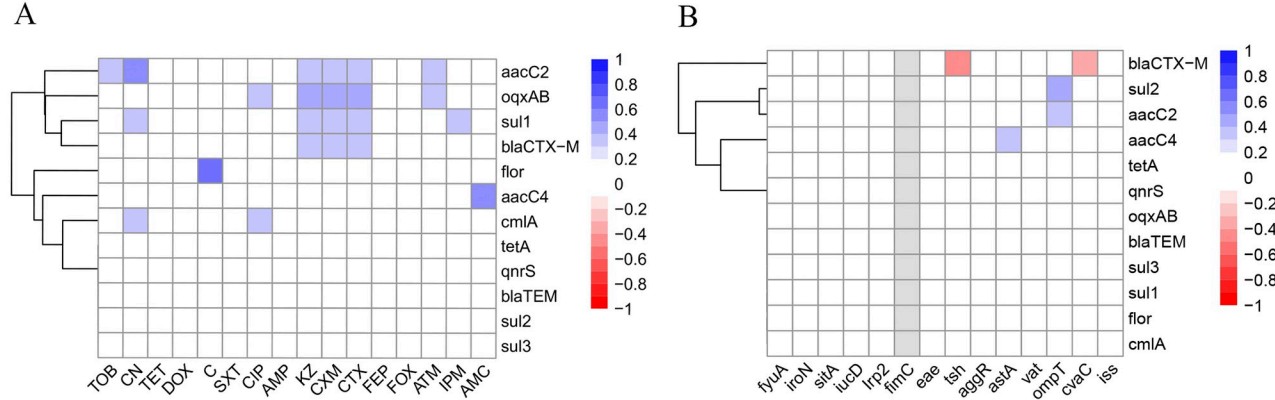

**Fig 3. Heatmap of the correlation-coefficient (r) between ARGs and AMR or VAGs in 74 MDR *E. coli* strains from pet dogs.** Blue indicates positive association ($r > 0$, $P < 0.05$) and red indicates negative association ($r < 0$, $P < 0.05$). The color scale on the right of figure indicates the *r*-valve: (A) Heatmap of the correlation coefficient between ARGs and AMRs. The color scale and corresponding *r*-valve indicate the association between corresponding abscissa AMRs and ordinate ARGs. Twenty-three positive association pairs were observed, of which the strongest association was found between *flor* and C; (B) Heatmap of correlation coefficient between ARGs and VAGs. The color scale and corresponding *r*-valve indicate the association between corresponding VAGs (abscissa) and ARGs (ordinate). Five association pairs were observed (3 association pairs were positive and 2 association pairs were negative), of which the strongest positive association was found between *ompT* and *sul2*, the strongest negative association was found between *bla*CTX-M and *tsh*.

9.46% strains belonged to commensal group (B1) ($P < 0.001$). Furthermore, we analyzed the average number of VAGs carried by per strain in different groups, of which the highest observed was 8.28 in group B2, followed by 7.29 in group B1 and 5.57 in group D ($P < 0.01$). And no significant difference ($P > 0.05$) was observed in the average number of VAGs carried by per strain between commensal group (B1) and virulent extraintestinal-related groups (B2/D) (Fig 4B).

In addition, 42 different sequence types (STs) were identified in 74 MDR *E. coli* isolates (Fig 4C). Eleven strains belonged to ST10, followed by ST155 (4 strains), ST162 (4 strains), ST457 (4 strains), ST127 (3 strains) and ST410 (3 strains), the remaining 36 STs contained only one strains, respectively. Moreover, two new sequence types (nST1, nST2) were identified in our present study (S4 Table). By using goeBURST algorithm, 42 STs were clustered into 6 clonal complexes (CCs) and 23 singletons. Among the 6 clonal complexes, ST44, ST175, ST744, ST1415 and ST2197 were included in clonal complex 10CC with the founder ST10 (ST-10CC), and ST-10CC was the predominant lineage containing 17 MDR strains. ST345, ST410 and ST423 belonged to 90CC with the founder ST90 (ST-90CC); ST58, ST155 and nST1 belonged to 155CC with founder ST155 (ST-155CC); ST6209 and ST156 belonged to one clonal complex with the founder ST156 (ST-156CC), ST9580 and ST206 belonged to one clonal complex with the founder ST206 (ST-206CC), respectively. However, the CC contained ST359 and ST101 without any founder ST (Fig 4D). The heatmap of phylogroups and MLST distribution showed that 11 (14.86%, 11/74) strains simultaneously belonged to B2 group and ST10, which were the most prevalent clones in our present study (Fig 4E).

## 4. Discussion

The widespread use of antibiotics has significantly caused the increase of MDR *E. coli* which were isolated from companion animals [33]. To understand the characteristics of MDR strains and evaluate the potential threat, we detected the AMR of 135 *E. coli* isolates from pet dogs and further screened ARGs, VAGs, phylogenetic grouping and MLST for 74 MDR strains.

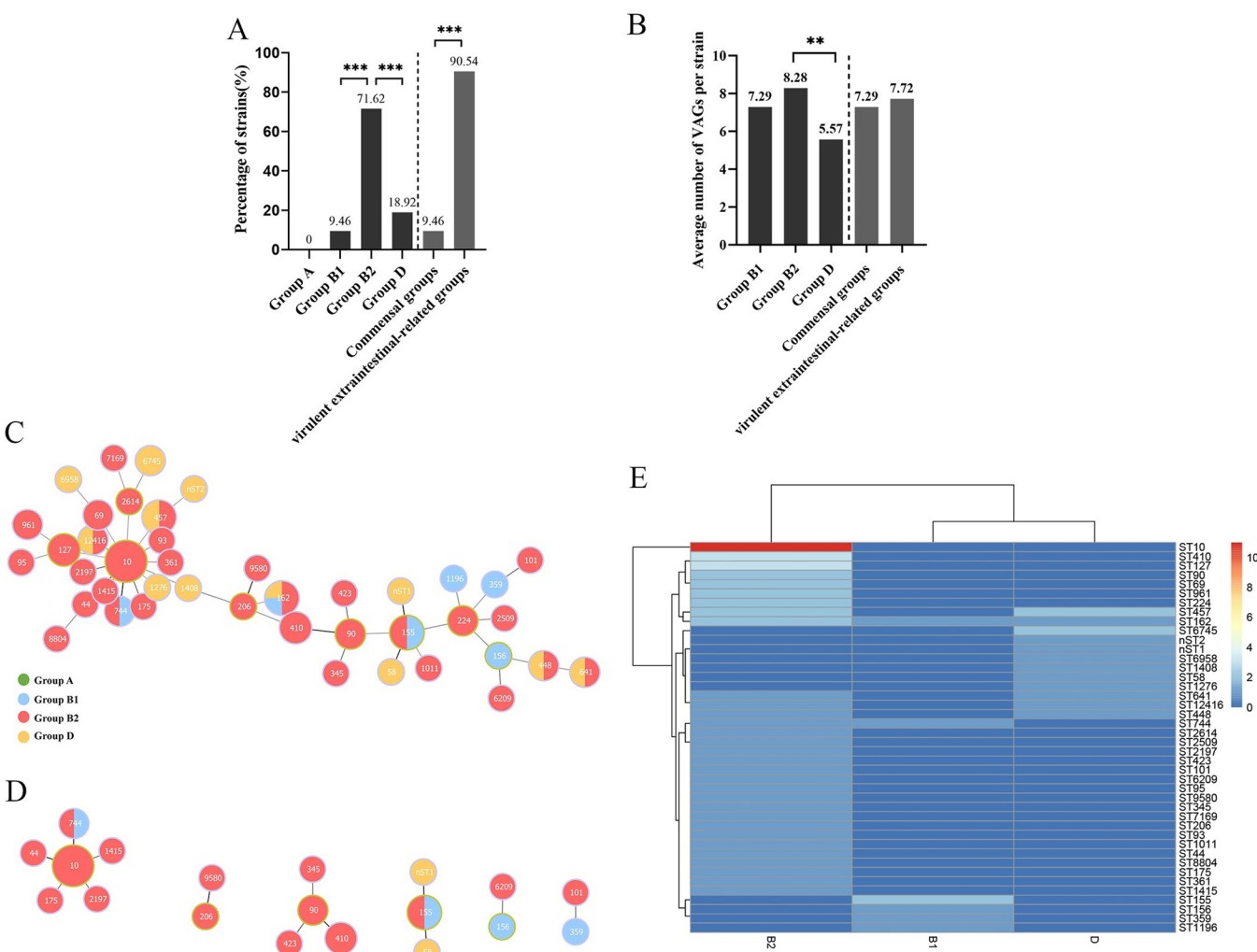

**Fig 4. Distribution of phylogenetic groups and STs in 74 MDR *E. coli* strains from pet dogs.** (A) The distribution of phylogroups in MDR strains. Commensal groups included groups A and B1, and virulent extraintestinal-related groups included groups B2 and D. Marking * represents a significant difference, *, $P < 0.05$; **, $P < 0.01$; ***, $P < 0.001$. Significantly, the virulent groups were the most prevalent; (B) The average number of VAGs per isolate in each phylogroup. Marking * represents significant difference, *, $P < 0.05$; **, $P < 0.01$; ***, $P < 0.001$. Obviously, the highest value observed was 8.28 in group B2.; (C) Minimum spanning tree of MLST types in MDR strains. The circle size indicates the proportion of isolates belonging to the ST. The color within each circle represents phylogroups and indicates the proportion of isolates belonging to different phylogroups. Each link between circles indicates one mutational event and the distance is scaled as the number of allele differences between STs. The yellow-green outlines of the circles represent the ST is the founder ST of one clonal complex (CC), and the other STs (with purple outlines of the circles) of the CC are derived from the founder ST with two allelic differences. The 74 MDR strains exhibited a high diversity of STs (42 STs were identified) in our present study and ST10 was the most prevalent; (D) The clonal complexes (CCs) among the 42 STs. Forty-two STs were clustered into 6 clonal complexes, and the remaining 23 STs were single. ST-10CC was the most prevalent lineage containing 6 STs; (E) Heatmap demonstrates the distribution of STs and phylogroups. The color scale and corresponding value indicate the number of *E. coli* isolates belonging to the corresponding phylogroups (abscissa) and STs (ordinate). Blue indicates a low number of strains, white indicates an intermediate value, and red indicates a high number of strains. As the heatmap shows, B2-ST10 was the most prevalent clone.

One hundred and thirty-five *E. coli* isolates from pet dogs showed high resistance to TET, AMP and SXT, which was consistent with another study in China [34]. Notably, TET has been used for treating animal infection disease and has also been used as "growth promoters" or "feed efficiency products" for a long time [35,36], which may activate the resistance of TET in *E. coli*. Once the antimicrobial-resistant *E. coli* occurs, it exists for a long time due to hard elimination [37]. In addition, TET resistance has been widely observed in *E. coli* from animals, such as dogs [13,24], yaks [38] and pigs [39], and the antibiotic resistance genes encoding

AMR can be transmitted through environment and mobile genetic elements [40–42], which may be one of the reasons for high TET-resistant *E. coli* detected in our present study, even though TET has not been used in the area recently. The similar phenomenon was also found in the resistant strains to 3rd/4th generation cephalosporins, fluoroquinolones and carbapenems antibiotics in our present study, which have been advocated to avoid or restrict use in veterinary antimicrobial selection in China [34,43]. The resistance rates of AMP and SXT were consistent with the results from pet dogs in Nigeria, Australia and Brazil [19,23,44]. According to previous studies, frequent use of antimicrobial agents in clinics may be responsible for the high resistance to AMP and SXT [45,46], the β-lactam antibiotics have also been widely used in the location of the present study, and SXT is one of the most animal-used antimicrobials in China [45]. In our study, we further screened 74 MDR strains from 135 *E. coli* isolates. Compared with previous studies in Sichuan, the detection rate of MDR strains among *E. coli* isolates from pet dogs decreased from 100% in 2017 [47] and 70% in 2018 [48] to 54.81% in our study during 2021–2022, which was similar to the trend observed in Northeast China (which decreased from 76.92% in 2012–2013 to 62.42% in 2021) [43]. The decrease in the MDR detection rate may be due to that the relevant authorities of China have released several policies to curb the increase in antimicrobial resistance in animal-derived bacteria [43]. Although the detection rate of MDR *E. coli* from pet dogs in Sichuan has decreased, it is still higher than that detected in other countries [44,49]. Overall, resistant strains may have been circulating in the dog population long before the antibiotic usage restriction were introduced, and AMR strains are currently difficult to eliminate. The high detection rates of MDR *E. coli* in our study implied that more effective measures should be taken to control the occurrence and spread of MDR *E. coli* from companion animals.

It is well-known that the antibiotic resistant phenotype of bacteria is related to ARGs [50]. Therefore, we further analyzed the distribution of ARGs in 74 MDR strains. In our study, 12 out of 18 ARGs were detected in 74 MDR strains (detection rates ranged from 16.22% to 95.95%), of which *tetA*, *bla*TEM and *bla*CTX-M were the dominant ARGs. Notably, the detection rate of *tetA* in our study was higher than previous studies from dogs in China which ranged from 28% to 85.08% [43,51,52]. The *bla*TEM and *bla*CTX-M were the prevalent β-lactam resistance genes in our study which were consistent with previous study for *E. coli* from dogs in China [43]. We further analyzed the prevalent subtypes of *bla*CTX-M genes, *bla*CTX-M-55 (in *bla*CTX-M-1 group) and *bla*CTX-M-14 (in *bla*CTX-M-9 group) were more prevalent. Among variants of *bla*CTX-M-9 group, *bla*CTX-M-14 has been considered as a common CTX-M variant worldwide, especially in China, Korea and Japan [53]. A high prevalence of *bla*CTX-M-14 was also observed in our present study (79.10%). Notably, *bla*CTX-M-14 was usually found on plasmid (such as IncF and IncK) according to previous studies [54,55]. The high prevalence of *bla*CTX-M-14-positive strains observed in our present study indicated a high risk of potential transmission. For variants of the *bla*CTX-M-1 group, *bla*CTX-M-55 has become the second most common CTX-M subtype in Chinese clinical *E. coli* isolates after *bla*CTX-M-14 [54], which was also dominant in our present study. Furthermore, Cottell et al. has observed the frequent clonal transmission of *bla*CTX-M-55-positive *E. coli* between different hosts (humans and animals such as duck, chicken, and swine) [54]. The high prevalence of *bla*CTX-M-14-positive and *bla*CTX-M-55-positive strains observed in our present study suggests more studies should focus on the capability of clonal transmission between different hosts in the future.

Increasing studies proposed that the relationship between AMRs and ARGs is not completely consistent [22,56]. The detection rates of β-lactam and tetracycline antibiotics resistant phenotypes were generally consistent with the related ARGs. However, for sulfonamides, aminoglycosides, amide alcohols, and quinolones, the detection rates of related ARGs were fluctuated, e.g., the resistant rate of sulfonamides was 90.54% in MDR strains, while the

detection rates of *sul* ranged from 17.57% to 70.27%. Moreover, we further analyzed the statistical associations between ARGs and AMRs, 23 positive association pairs were observed in MDR strains. Only *flor* and C, *bla*$_{CTX-M}$ and CTX/CXM/KZ, *oqxAB* and CIP, *aacC2* and TOB/CN showed consistency in ARG and related AMR. The other 16 association pairs were not completely consistent (Fig 3A). The similar phenomenon was also found in *E. coli* from captive non-human primates [22] and waterfowl [56] in China which indicated the different expression of ARGs, such as the abnormal expression of ARGs and the expression of ARGs has not reached that level which can activate the antibiotic resistance, may be a possible reason the incomplete consistency between AMR and ARGs [56].

Recent studies have shown that there was an association between ARGs and VAGs, and have proposed that they were linked and interacted with each other [6,56]. In order to explore the virulence traits and the association of ARGs and VAGs in MDR *E. coli*, we analyzed the distribution of VAGs in MDR strains. In our present study, 25 VAGs were selected for detection according to previous studies, among which 12 VAGs (*eae, papA, bfpA, aggR, pic, astA, ipaH, stx1, stx2, elt, esta and estb*) are related to Diarrhoeagenic *Escherichia coli* (DEC) [57–59], the remaining 13 VAGs (*fyuA, iroN, sitA, iucD, Irp2, fimC, tsh, vat, ompT, cvaC, iss, hlyF, hlyA*) are related to Extraintestinal *Escherichia coli* (ExPEC) [60–62]. Eleven ExPEC-related VAGs and 3 DEC-related VAGs (*eae, astA, aggR*) were detected in our present study. And the detection rates of VAGs in our study, which ranged from 4.99% to 100%, were higher than previous studies in dogs from Shaanxi and Shandong, China (ranging from 0.6% to 81.6% in Shaanxi and 2.53% to 87.34% in Shandong, respectively) [34,63]. Moreover, 75.68% (56/74) of the MDR strains simultaneously combined at least one ExPEC-related and DEC-related VAG, indicating the MDR strains in our present study harbored VAGs related to different pathotypes and could be considered as potentially virulent hybrid pathogenic strains. MDR strains with potential hybrid pathogen detected in pet dogs will pose a threat to other companion animals and their owners.

In addition, the association between VAGs and ARGs among MDR strains was further analyzed. Five associated pairs were observed, of which 3 pairs were positive and 2 pairs were negative. Compared with other studies of *E. coli* from giant pandas (46 associated pairs, of which 45 pairs were positive and 1 pair was negative) [6] and waterfowl (43 associated pairs, of which 36 pairs were positive and 7 pairs were negative) [56], the associations observed between ARGs and VAGs in MDR strains from pet dogs were not common in our present study. The possible reasons for drawing associations between VAGs and ARGs may be related to the co-location on the same mobile genetic elements, while negative associations indicated gene incompatibilities [64]. Recent studies also showed that many ARGs were inserted into conjugative plasmids carrying VAGs, which may provide conditions for drawing positive associations between ARGs and VAGs [65]. In our study, the strongest positive association was observed between *ompT* and *sul2*, which is usually detected in plasmids, indicating the positive association between *ompT* and *sul2* may be related to the plasmid [40,66]. Similarly, the location of *aacC*, which usually was detected in the variable region of integron, may be related to the other two positive associations (*aacC2/ompT, aacC4/astA*) observed in our present study [5]. The positive association between ARGs and VAGs implied the high risk of plasmid-mediated co-transmission of ARGs and VAGs in *E. coli*, which could accelerate the spread of VAGs and ARGs within *E. coli* populations and enhance the emergence possibility of new pathogens with increased virulence and resistance potential. Negative associations observed in our present study may also be related to mobile genetic elements. For example, the VAG *cvaC* is usually found in virulent IncF plasmids [66], while *bla*$_{CTX-M}$ is usually found in integron [67], the different locations on mobile genetic elements may be related to the negative associations between *bla*$_{CTX-M}$ and *cvaC* in our present study. While the specific mechanism for the associations observed in our study required more comprehensive studies to validate.

Previous studies have shown that groups A and B1 are considered to be commensal groups, and groups B2 and D are considered to be virulent extraintestinal-related groups [27], which have been classified as potentially pathogenic [5,6]. A study from Iran showed that resistant *E. coli* strains mostly belonged to groups B2 and D [68]. Similarly, our present results showed a high prevalence of strains belonging to virulent extraintestinal-related groups (B2 and D) were detected among MDR *E. coli* from pet dogs. According to previous study, phylogenetic grouping is closely related to virulence genes (group A and B1 strains carry fewer virulence genes, while group B2 and D strains carry more virulence genes) [6], whereas in our present study, no significant difference was observed in the average number of VAGs carried by per strains between commensal groups and virulent extraintestinal-related groups. These phenomena suggest that antimicrobial resistance may increase the possibility of carrying more virulent factors (for example, the number of VAGs carried) in *E. coli* strains [69], implying that highly antibiotics resistant intestinal *E. coli* strains may also lead to extraintestinal infections which could be an emergent public health issue for humans, animals and the environment [70,71].

At last, we used MLST to analyze the dominant lineage of MDR strains, and our results showed that 74 MDR strains exhibited a high population diversity of STs (42 STs were identified), of which ST10 was the most prevalent. Another study of *E. coli* from dogs in China has found that the pandemic clone was ST131 (accounting for 9.8%) [63], which was identified as the most common ST in ExPEC, covering all geographical regions [72]. Moreover, studies from Australia and Thailand showed that ST354 and ST410 were the dominant clones in *E. coli* isolates from dogs, respectively [15,16]. In our study, the most common ST was ST10 (occurring in 14.86% of strains) which was different from the above studies, indicating that the prevalent ST lineage of *E. coli* isolates from pet dogs may vary in different geographical regions. ST10 was widely detected in *E. coli* from other animals in China, such as pigs [73] and chickens [74], indicating that ST10 may be the prevalent ST in *E. coli* strains from animals in China. Moreover, ST10 strains mostly belonged to B2 groups in our present study, and this phenomenon was also observed in *E. coli* isolates from chickens [74] and in the ExPEC isolates from humans [75]. In other studies, ST10 strains identified from pigs [73] and humans [76] mostly belonged to group A or D. This phenomenon indicated that the relationship between ST and phylogroups may vary in *E. coli* isolated from different hosts.

## 5. Conclusions and limitations

Seventy-four MDR strains were observed among 135 *E. coli* isolates from pet dogs, of which 12 ARGs and 14 VAGs were detected, implying that the *E. coli* isolates can be considered as a pool of ARGs and VAGs. Moreover, groups B2 and D, which are classified as potentially pathogenic, were predominant in the MDR isolates, and ST10 was the prevalent clonal lineage. Our findings have important implications for a preliminary understanding of the characteristics of MDR *E. coli* from diarrheal dogs and evaluating the potential risk of resistance and virulence in canine *E. coli*.

In our present study, the number of samples included was limited and the sampling site was only at our Veterinary Teaching Hospital. Future investigations should include more samples and expand sampling areas. Moreover, the use of High-throughput Sequencing or Whole Genome Sequencing could provide more comprehensive information on MDR *E. coli* strains from diarrheal dogs.

## Supporting information

**S1 Table. PCR primer and conditions for antibiotic resistance genes (ARGs) used in this study.**
(XLSX)

**S2 Table. PCR primer and conditions for virulence-associated genes (VAGs) used in this study.**
(XLSX)

**S3 Table. PCR primer and conditions of 7 housekeeping genes for MLST and 3 related genes for phylogroups used in this study.**
(XLSX)

**S4 Table. Distribution of sequence types (STs) in MDR *E. coli* strains from pet dogs (n = 74).**
(XLSX)

**S5 Table. Details of STs, phylogroups, AMR, ARGs and VAGs pattern for MDR *E. coli* isolated from pet dogs.**
(XLSX)

**S6 Table. The performance Standards for Antimicrobial Susceptibility testing in the present study.**
(XLSX)

**S7 Table. Clinical history of the pet dogs which successfully isolated *E. coli* strains in our present study.**
(XLSX)

## Acknowledgments

We thank Dr Ziyao Zhou, Junai Gan, and Lei Deng for the English language revision.

## Author Contributions

**Conceptualization:** Xiaoli Zhang, Keyun Shi, Zhijun Zhong.

**Data curation:** Yu Yuan, Yan Hu, Wenhao Zhong, Zhijun Zhong.

**Investigation:** Yu Yuan, Wenhao Zhong, Shulei Pan.

**Methodology:** Xiaoli Zhang, Keyun Shi, Zhijun Zhong.

**Software:** Xiaoli Zhang, Guangneng Peng, Ya Wang, Qigui Yan, Keyun Shi, Zhijun Zhong.

**Supervision:** Liqin Wang, Haifeng Liu, Shaqiu Zhang.

**Validation:** Guangneng Peng, Ya Wang, Qigui Yan.

**Visualization:** Yu Yuan, Wenhao Zhong, Shulei Pan.

**Writing – original draft:** Yu Yuan, Yan Hu, Zhijun Zhong.

**Writing – review & editing:** Ziyao Zhou, Yan Luo, Zhijun Zhong.

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
