## [Decision Letter · Decision Letter 0]

16 Nov 2023

PONE-D-23-30998Characteristics of MDR E. coli strains isolated from Pet Dogs with clinic diarrheal: A Pool of Antibiotic Resistance Genes and Virulence-Associated GenesPLOS ONE

Dear Dr. Zhong,

Thank you for submitting your manuscript to PLOS ONE. After careful consideration, we feel that it has merit but does not fully meet PLOS ONE’s publication criteria as it currently stands. Therefore, we invite you to submit a revised version of the manuscript that addresses the points raised during the review process.

We look forward to receiving your revised manuscript.

Kind regards,

Md. Tanvir Rahman, DVM, MSc, PhD

Academic Editor

PLOS ONE

Journal Requirements:

"This research was funded by the National Key Research and Development Program of China (2018YFD0500900, 2016YFD0501009), the Chengdu Giant Panda Breeding Research Foundation (CPF2017-05, CPF2015-4) and the Science and Technology Achievements Transfer Project in Sichuan province (2022JDZH0026)."

4. Please remove your figures from within your manuscript file, leaving only the individual TIFF/EPS image files, uploaded separately. These will be automatically included in the reviewers’ PDF.

Additional Editor Comments:

Dear authors,

Please see the comments of the reviewers and the try to address them properly.

Reviewers' comments:

Reviewer's Responses to Questions

**Comments to the Author**

1. Is the manuscript technically sound, and do the data support the conclusions?

Reviewer #1: Partly

Reviewer #2: No

Reviewer #3: No

2. Has the statistical analysis been performed appropriately and rigorously? 

Reviewer #1: I Don't Know

Reviewer #2: Yes

Reviewer #3: I Don't Know

3. Have the authors made all data underlying the findings in their manuscript fully available?

Reviewer #1: Yes

Reviewer #2: No

Reviewer #3: Yes

4. Is the manuscript presented in an intelligible fashion and written in standard English?

Reviewer #1: No

Reviewer #2: No

Reviewer #3: No

5. Review Comments to the Author

Reviewer #1: General:

The study by Yuan et al. did screen 135 E. coli obtained from the feces of diarrheic pet dogs for their antimicrobial resistance, the antibiotic resistance genes und virulence associated genes, conducted phylogenetic grouping and multi locus sequence typing.

As the study serves the purpose characterising E. coli strains in order to monitor its prevalence and distribution, I strongly advise the authors to work with a writing coach or copyeditor to improve the language and readability.

The authors need to rerun the phylogenetic grouping and/or MLST, as it is highly unlikable that ST10 belongs to phylogroup B2. Please recheck!

The authors should reexamine the references cited in the discussion to determine whether the comparisons thus drawn are indeed based on comparable study conditions.

The authors should reconsider, why they are drawing associations between VAGs and ARGs and explain the importance of these results.

Please add limitations of the study.

Specifics:

Line 52-53: The authors should clarify the statement. Reference 11 was unable to find link between the examined resistance and virulence genes.

Line 58: The authors should revise the use of the term „virulent groups“ (B2 and D), as Clermont et al (Reference 19) refer to virulent extra-intestinal strains belonging to groups B2 and D. Please check throughout the manuscript.

Line 58, 216, 224, 304, 342, 360: Italics in the scientific name

Line 61-63: The authors should clarify the sentence to avoid confusion.

Line 70-71: Bulletin of the Ministry of Agriculture and Rural People’s Republic of China, 2020). (27). Please check punctuation.

Line 76: Please change diarrheal in diarrheal disease or diarrhea.

Line 99: Please change 16 S rRNA into 16S rRNA and Primer:5’-G to Primer: 5’-G

Line 114, 173, 332, 347, 349: Please use the correct abbreviation for SXT.

Line 116: Since there are no breakpoints for tobramycin, ciprofloxacin, cefuroxime,

cefotaxime, cefepime, cefoxitin, aztreonam in CLSI Vet06 2023, the authors should clarify how they obtained the breakpoints.

Line 138-140: The authors should explain why they used the protocol of Clermont from 2000 which distinguishes phylogroups A, B1, B2, and D instead of the new protocol from 2013 which enables an E. coli isolate to be assigned to one of the eight phylogroups (A, B1, B2, C, D, E, F and Escherichia cryptic clade I).

Line 177-178: Please change Results s howed to Results showed

Line 179-180: Please change were o bserved to were observed

Figure 1.B: Please change the image in favor of clarity and quality. For example, summarize combinations that occur only once and enlarge the remaining that occur multiple times.

Line 186: Please change 0 R to 0R according to figure 1.A. Please revise the language in the following sentences

Table 1: Please change ”Antibiotics” in the top heading to “Antibiotic”. Please insert a space between number of isolates and percentage in the first column. Please change 1rd/2th to 1st/2nd. As the interpretation of zone diameter or MIC changes with revised breakpoints every few years, a table with the results of the zone diameters should be added as a supplementary figure.

Line 201, 212, 206, 207, 373: Please check gene names.

Line 2018-220: The authors should clarify the sentence to avoid confusion.

Table 2: Please remove the % from the percentage of resistance to beta-lactams. Your results of tetracyclines suggest you found strains which exhibited resistance to doxycycline and at the same time susceptibility to tetracycline. Did you recheck these?

Line 261: You write “The other 2 pairs (blaCTX-M/tsh, blaCTX-M/iss) were negative association (r < 0, P < 0.05)”. In Figure 3 it shows a negative association with cvaC instead of iss. Please check.

Line 284: Please check sentence with keeping in mind that no strains belonged to phylogroup A.

Line 286-288: Please explain the marking representing significance in Figure 1.B.

Figure 4.C and 4.D: Please clarify the meaning of the different coloured outlines of the circles.

Figure 4.E: Please clarify on which data the tree is based.

Line 368-369: You stated you found blaTEM-1, blaTEM-135, and blaTEM-176. Neither of these is a ESBL-encoding gene. Please rephrase the sentence accordingly.

Line379-380: Please add in Material and Methods, how you located the blaCTX-M-14 on the plasmid and present the data in results.

Line 383-388: The authors use very general terms in the two sentences. Please clarify, what “paying more attention” means. Please explain in more detail what you mean by “diversity and evolution” and the implications of that, or refrain from using the sentence.

Line 399-400: The authors are repeating the sentence from 391-392. Please amend.

Line 405-406: The authors speak in very general terms. The manuscript would improve by the use of examples.

Line 412: Please cross-reference the percentages presented with the sources.

Line 412-414: As in the cited study a different set of virulence genes was tested, a comparison with the results of the study appears difficult. Please revise or delete.

Line 416-419: Since the study does not provide data on the prevalence of combining VAGs separated into categories of ExPEC-related and DEC-related VAGs, it is difficult to follow the argument of the sentence.

Line: 428-429: The authors repeat the results already given in discussion. Please delete.

Line 431-434: The authors explain the possible reason for the positive correlation between sul2 and ompT very well. Is there a similar rationale for the other two positive pairs? Is anything known about possible reasons for the negative association pairs?

Line 442-443: Please clarify the sentence to avoid the reader’s confusion.

Line 448-449: Please clarify the sentence to avoid the reader’s confusion.

Line 452: According to the reference, ST131 has been identified the most common sequence type in ExPEC. Please rephrase.

Line 455: The reader might wonder as ST10 is the most prevalent sequence type in the study, if there is anything known about its distribution?

Line 462-463: Please specify.

Reviewer #2: (1) Abstract: So far I know, any abstract has four essential parts with few additional or optional parts, such as objective(s), methodology, findings, and concluding remarks as essential parts and background, limitations, future indication etc could be present as additional portion(s). In this abstract, the objectives and methodology were not properly written, even absent. The abstract should be re-written properly, concisely and with proper structures.

(2) Introduction: In my opinion, research gaps, justification and novelty of the present research were not mentioned or presented properly.

(3) Methodology:

(A) Sample size could be larger, and could be determined using formula. However, the less number of samples could be explained properly in the 2.1 section.

(B) Line 77: why this findings unnecessarily came here?

Line 80-81: Could you please provide the permission number?

(C) strain isolation portion (section 2.2) was made unnecessarily complex.

(D) Line 113-115: these information could be added in discussion.

(E) Line 124-125: please rephrase the sentence.

(F) Line 127-130: Could be added to the discussion.

(G) The 2.4 section was not described properly and scientifically. Supplementary Table information should be mentioned.

(4) Results: poorly represented.

(A) for determination of ARGs and AMR or VAGs association, in my opinion, all the E. coli isolates should be taken into consideration. Only MDR or resistant isolates can not give any conclusive data. Moreover, for this purpose, the sample size should be larger than the present research work.

(B) Phylogroup and ST distribution was not properly depicted.

(C) Text was elaborately presented, so that the table 1 and 2 were unnecessarily added.

(5) Discussion: poorly written. repetition of results came many times. Exact platform was not made.

(6) Conclusions: did not reflect the research outcome properly.

(7) References: huge reference list. should be restricted within 50-55. better less than 40.

(8) Others: Grammatical errors and poor English in many places of the manuscript.

Moreover, I could not find any new information from this research works. Though have few new ideas. However, following published articles could be mentioned as justification or research gaps...

https://doi.org/10.1128%2FJCM.40.10.3586-3595.2002

Sanchez S, McCrackin Stevenson MA, Hudson CR, Maier M, Buffington T, Dam Q, Maurer JJ. Characterization of multidrug-resistant Escherichia coli isolates associated with nosocomial infections in dogs. Journal of Clinical Microbiology. 2002 Oct;40(10):3586-95.

https://doi.org/10.5455/javar.2020.g466

Deb, P., Das, T., Nath, C., Ahad, A., & Chakraborty, P. (2020). Isolation of multidrug-resistant Escherichia coli, Staphylococcus spp., and Streptococcus spp. from dogs in Chattogram Metropolitan Area, Bangladesh. Journal of Advanced Veterinary and Animal Research, 7(4), 669-677

https://doi.org/10.1128/AEM.00599-11

Karczmarczyk M, Abbott Y, Walsh C, Leonard N, Fanning S. Characterization of multidrug-resistant Escherichia coli isolates from animals presenting at a university veterinary hospital. Applied and Environmental Microbiology. 2011 Oct 15;77(20):7104-12.

https://doi.org/10.3389/fvets.2023.1104812

Tong YC, Zhang YN, Li PC, Cao YL, Ding DZ, Yang Y, Lin QY, Gao YN, Sun SQ, Fan YP, Liu YQ. Detection of antibiotic-resistant canine origin Escherichia coli and the synergistic effect of magnolol in reducing the resistance of multidrug-resistant Escherichia coli. Frontiers in Veterinary Science. 2023 Mar 15;10:1104812.

https://doi.org/10.1016/j.prevetmed.2022.105767

Osman M, Albarracin B, Altier C, Gröhn YT, Cazer C. Antimicrobial resistance trends among canine Escherichia coli isolated at a New York veterinary diagnostic laboratory between 2007 and 2020. Preventive Veterinary Medicine. 2022 Nov 1;208:105767.

Reviewer #3: The manuscript described isolation of MDR E. coli strains from pet dogs carrying different antimicrobial resistance genes. The isolates were mostly associated with virulent phylogroups (B2 and D) and ST10 clonal lineage. The objectives are interesting and the methodology is standard. Language is poor and the manuscript is full of typing, grammatical and syntax errors which should be rectified by a well-versed English speaker. Most of the findings lack proper justification in the discussion section. Consequently, the conclusions are not supported with the findings.

Following issues should be replied point-wise.

Line 28. blaTEM-1 cannot be considered as ESBL…if it is necessary to describe TEM-1, please replace the ‘ESBL’ with ‘beta-lactamase’ throughout the manuscript

Line 76. ‘clinical diarrheal’ should be replaced with ‘clinical diarrhea’

Line 81. ‘animal ethics committee’ should be added

Line 82. ‘professional veterinaries’ should be replaced with ‘professional veterinarians’

Line 87. ‘Eppendorf (EP) tubes’ should be replaced with ‘micro centrifuge tubes (Eppendorf)’

Line 87. Clinical history of the pet dogs is missing, age/breed/sex of the diarrhoeic dogs, types of diarrhoea (mucoid/ bloody etc.), description of other clinical symptoms like fever, anorexia, vomition etc. should be added

Line 91. ‘Fecal samples were enriched in Luria-Bertani (LB) broth at 37 °C, 120 r/min for 24 h’…if the authors mean to describe about incubation in shaking incubator, that should be indicated

Line 94. ‘singe clone’ ??

Line 113. I agree with the authors that the canine E. coli isolates earlier showed resistance against the antibiotic discs selected in the present study. However, are these antibiotics commonly used in pet/companion animals in the study location? Did you consider to go through the prescriptions or did you conduct any qualitative survey to know the antibiotic usage pattern in local canine population?

Line 135. After BLASTing did you submit the sequences into genbank? If so, please indicate the accession numbers

Line 334. Did you find use of tetracycline as growth promoter in the study location? Moreover, how did you correlate growth promotional use of tetracyclines with higher occurrence of TET-resistant isolates in canine population?

Line 345. ‘Those phenomena indicated that AMR sustain for a long time and not easy to eliminate, once AMR is activated’ ….I found the sentence ambiguous which should be modified or deleted. It seems that implementation of antibiotic use restriction in the study location is not very effective or there are resistant strains circulating in the canine population since long before the imposition of antibiotic usage restriction and the strains are difficult to eliminate currently.

Line 348. ‘Frequent use of β-lactams and sulfonamides antimicrobial agents in clinic…’ another speculation, it should be justified with the screening of prescriptions in the clinics used in the present study

6. PLOS authors have the option to publish the peer review history of their article (what does this mean?). If published, this will include your full peer review and any attached files.

Reviewer #1: No

Reviewer #2: No

Reviewer #3: No

---

## [Author Response · Author response to Decision Letter 0]

15 Jan 2024

Dear Editor:

Thank you very much for your letter and the reviewers’ comments regarding our manuscript submitted to “PLOS ONE” (Manuscript ID: PONE-D-23-30998). We have checked the article and revised it according to the comments, and carefully proof-read the manuscript. The language presentation was improved with assistance from English speakers with appropriate research background. All revisions were marked in revised manuscript with track changes. 

We hope, with these modifications and improvements based on your suggestion and the reviewer’s comments, the quality of our manuscript would meet the publication standard of “PLOS ONE”. Once again, we acknowledge your comments and constructive suggestions very much, which are valuable in improving the quality of our manuscript.

We submit here the revised manuscript with track changes as well as a list of changes. Thanks again for your reconsideration of our manuscript for publication in your journal. If you have any question about this paper, please don’t hesitate to let me know.

Best wishes for you!

Sincerely yours,

Zhijun Zhong 

College of Veterinary Medicine

Key Laboratory of Animal Disease and Human Health of Sichuan Province

Sichuan Agricultural University

Chengdu 611130, P. R. China

E-mail: zhongzhijun488@126.com

Response to Reviewer 1 Comments

Response to Reviewer:

 Thank you very much for your time and thoughtful comments, many of which have been incorporated into the revised manuscript (Tracked Version). Detailed responses are as follows.

Reviewer #1: 

General:

The study by Yuan et al. did screen 135 E. coli obtained from the feces of diarrheic pet dogs for their antimicrobial resistance, the antibiotic resistance genes und virulence associated genes, conducted phylogenetic grouping and multi locus sequence typing.

Comment 1：As the study serves the purpose characterising E. coli strains in order to monitor its prevalence and distribution, I strongly advise the authors to work with a writing coach or copyeditor to improve the language and readability.

Response 1：Thanks for your comments. We have carefully checked the English expression and made extensive English revisions under the scrutiny of native English speakers with appropriate research background.

Comment 2：The authors need to rerun the phylogenetic grouping and/or MLST, as it is highly unlikable that ST10 belongs to phylogroup B2. Please recheck!

Response 2：Thanks for your comments. We reran the phylogenetic grouping and MLST to check our results carefully, and our results were correct. Actually, this phenomenon was also observed in E. coli isolates from chickens in China (1) and the ExPEC isolates from humans in Turkey (2). Previous studies showed that ST10 strains identified in E. coli from pigs in China (3) and humans in Spain (4) were mostly belonged to A or D group. These results indicates that the relationship between ST and phylogroups may vary in different E. coli isolated from different hosts. We have also added the corresponding discussion content for this phenomenon in our revised manuscript (Tracked Version) (Line 593-598).

Commet 3：The authors should reexamine the references cited in the discussion to determine whether the comparisons thus drawn are indeed based on comparable study conditions.

Response 3：Thanks for your comments. We have carefully checked the references to ensure that the comparisons drawn are indeed based on comparable study conditions, and we have modified the references in our revised manuscript.

Comment 4: The authors should reconsider, why they are drawing associations between VAGs and ARGs and explain the importance of these results. 

Response 4：Thanks for your comments. we have added content in discussion for possible reasons of the associations between VAGs and ARGs and explanation of their importance in “discussion” section in our revised manuscript (Tracked Version) (Line 521-543): The possible reasons for drawing associations between VAGs and ARGs may be related to the co-location on the same mobile genetic elements, while negative associations indicated gene incompatibilities (5). Recent studies also showed that many ARGs were inserted into conjugative plasmids carrying VAGs (6), which also may provide conditions for drawing positive association between ARGs and VAGs. 

In our study, the strongest positive association was observed between ompT and sul2, which is usually detected in plasmids (7,8), indicating the positive association between ompT and sul2 may be related to the plasmid. The positive association between ARGs and VAGs implied the high risk of plasmid-mediated co-transmission of ARGs and VAGs in E. coli, which could accelerate the spread of VAGs and ARGs within E. coli populations and enhance the emergence possibility of new pathogens with increased virulence and resistance potential. 

Negative associations observed in our present study may also be related to mobile genetic elements. For example, the VAG cvaC usually be found in virulent IncF plasmids (9), while blaCTX-M is usually found in integron (10), the different location on mobile genetic elements may be related to the negative associations between blaCTX-M and cvaC in our present study. While the specific mechanism for associations observed in our present study required more comprehensive studies to validated.

Comment 5: Please add limitations of the study.

Response 5：Thanks for your comments. We have added “Limitations” in our revised manuscript (Line 612-616): In our present study, the number of samples included was limited and the sampling site was only at our Veterinary Teaching Hospital. Future investigations should include more samples and expand sampling areas. Moreover, the use of High-throughput sequencing or Whole Genome Sequencing could provide more comprehensive information on MDR E. coli strains from diarrheal dogs.

Specifics:

Line 52-53: The authors should clarify the statement. Reference 11 was unable to find link between the examined resistance and virulence genes.

Response: Thanks for your comments. We have deleted the reference in our revised manuscript.

Line 58: The authors should revise the use of the term „virulent groups “ (B2 and D), as Clermont et al (Reference 19) refer to virulent extra-intestinal strains belonging to groups B2 and D. Please check throughout the manuscript.

Response：Thanks for your comments, we have checked the whole manuscript and changed the “virulent groups” into “virulent extraintestinal-related groups” in our revised manuscript (Tracked Version) (Line 331, 554, 558, 569, and Line 945 in legends of Fig 4).

Line 58, 216, 224, 304, 342, 360: Italics in the scientific name

Response：Thanks for your comments, we have modified the italics in our revised manuscript (Tracked Version) (Line 66, 253, 428, line 923 in legends of Fig 2, line 943 in legends of Fig 4).

Line 61-63: The authors should clarify the sentence to avoid confusion.

Response：Thanks for your comments. We have modified the sentence in our revised manuscript (Tracked Version) (Line 68-75): Recent studies have identified a high population diversity of STs which related to zoonotic or pathotype in E. coli isolates from pet dogs, such as ST354, ST393, and ST457 E. coli were observed in companion animals from Australia. And the high-risk ExPEC clones associated with humans and multidrug resistance, including sequence type (ST) 38, ST131, ST224, ST167, ST354, ST410, ST617 and ST648, have been identified in cats and dogs in Thailand which suggested there may be clonal dissemination between pets and humans.

Line 70-71: Bulletin of the Ministry of Agriculture and Rural People’s Republic of China, 2020). (27). Please check punctuation.

Response：Thanks for your comments, we deleted the extra punctuation in our revised manuscript (Tracked Version) (Line 80-81).

Line 76: Please change diarrheal in diarrheal disease or diarrhea.

Response：Thanks for your comments, we have changed diarrheal into diarrhea in our revised manuscript (Tracked Version) (Line 96).

Line 99: Please change 16 S rRNA into 16S rRNA and Primer:5’-G to Primer: 5’-G

Response：Thanks for your comments, we have change 16 S rRNA into 16S rRNA and Primer:5’-G to Primer: 5’-G in our revised manuscript (Line 116-117).

Line 114, 173, 332, 347, 349: Please use the correct abbreviation for SXT.

Response：Thanks for your comments, we have checked the whole manuscript and modified the abbreviation for SXT in our revised manuscript (Tracked Version) (Line 142, 207, 382, and 409).

Line 116: Since there are no breakpoints for tobramycin, ciprofloxacin, cefuroxime,

cefotaxime, cefepime, cefoxitin, aztreonam in CLSI Vet06 2023, the authors should clarify how they obtained the breakpoints.

Response：Thanks for your comments. The references insert here is wrong. The correct referenced standard for our present study was the CLSI 2023 (Performance standards for antimicrobial susceptibility testing, M100-Ed33), and the Interpretive Categories and Zone Diameter Breakpoints for the antimicrobial agents which we tested are showed in Table S6. We have revised this reference in our revised manuscript.

Line 138-140: The authors should explain why they used the protocol of Clermont from 2000 which distinguishes phylogroups A, B1, B2, and D instead of the new protocol from 2013 which enables an E. coli isolate to be assigned to one of the eight phylogroups (A, B1, B2, C, D, E, F and Escherichia cryptic clade I). 

Response：Thanks for your comments. Firstly, the protocol of Clermont was still widespread used in recent studies of E. coli isolates from diarrheal dogs (11,12). Secondly, phylogenetic protocol from Clermont has shown that virulent extraintestinal strains mainly belong to groups B2 and D, whereas strains of groups A and B1 are usually devoid of extraintestinal virulence factors, indicating a link between phylogroups and extraintestinal virulence (13,14). Our present study focuses on the antibiotic resistance characteristics of MDR E. coli from diarrheal dogs and preliminary analysis of the phylogroups and MLST for MDR E. coli. Our results showed the most common ST was ST10 and ST10 strains mostly belonged to B2 groups. As the reviewer suggests that the new protocol from 2013 can get more comprehensive phylogroups, we will consider using the new protocol in our future research.

Line 177-178: Please change Results s howed to Results showed

Response：Thanks for your comments, the “showed” has been modified to “revealing “in our revised manuscript (Tracked Version) (Line 212).

Line 179-180: Please change were o bserved to were observed

Response：Thanks for your comments, we have changed o bserved into observed in our revised manuscript (Tracked Version) (Line 213).

Figure 1.B: Please change the image in favor of clarity and quality. For example, summarize combinations that occur only once and enlarge the remaining that occur multiple times.

Response：Thanks for your comments. We have added red boxes in the figure to highlight the prevalent resistant-phenotypes patterns (occur three times) in our revised Figure 1. And the other combinations (without red box) occur only once or twice. The corresponding explanation for the red boxes has been added to the figure legends of Fig 1.B (Line 919-921).

Line 186: Please change 0 R to 0R according to figure 1.A. Please revise the language in the following sentences

Table 1: Please change“Antibiotics”in the top heading to “Antibiotic”. Please insert a space between number of isolates and percentage in the first column. Please change 1rd/2th to 1st/2nd. As the interpretation of zone diameter or MIC changes with revised breakpoints every few years, a table with the results of the zone diameters should be added as a supplementary figure.

Response：Thanks for your comments. We have change 0 R to 0R according to figure 1.A (Line 914). For Table 1, we have changed“Antibiotics”in the top heading to “Antibiotic”, inserted a space between number of isolates and percentage in the first column, and change 1rd/2th to 1st/2nd (Line 228). Moreover, the zone diameter breakpoints for the antimicrobial susceptibility testing in our present study are showed in Table S6.

Line 201, 212, 206, 207, 373: Please check gene names.

Response：Thanks for your comments, we have modified the gene names in our revised manuscript (Tracked Version) (Line 235, 248, 239, 239, 443).

Line 218-220: The authors should clarify the sentence to avoid confusion.

Response：Thanks for your comments, we have modified the sentence to avoid confusion in our revised manuscript (Tracked Version) (Line 255-257): The average number of VAGs carried by per DEC-related strain (8.36) was significantly higher than that of DEC-unrelated VAGs strains (5.56) (P < 0.01).

Table 2: Please remove the % from the percentage of resistance to beta-lactams. Your results of tetracyclines suggest you found strains which exhibited resistance to doxycycline and at the same time susceptibility to tetracycline. Did you recheck these?

Response：Thanks for your comments, we have removed the % from the percentage of resistance to beta-lactams in our revised manuscript (Tracked Version) (Table 2). For the phenomenon of strains showing resistance to doxycycline and at the same time susceptible to tetracycline, we checked the detailed results of AMR, there are four strains resistant to doxycycline and at the same time susceptible to tetracycline. Moreover, the similar phenomenon was observed in our previous study for E. coli from captive non-human primates (15) and E. coli from captive giant pandas (16). While the reason for this phenomenon is unclear and needs to be investigated in further experiments.

Line 261: You write “The other 2 pairs (blaCTX-M/tsh, blaCTX-M/iss) were negative association (r < 0, P < 0.05)”. In Figure 3 it shows a negative association with cvaC instead of iss. Please check.

Response：Thanks for your comments, we have changed it in our revised manuscript (Line 247): The other 2 pairs (blaCTX-M/tsh, blaCTX-M/cvaC) were negative association.

Line 284: Please check sentence with keeping in mind that no strains belonged to phylogroup A.

Response：Thanks for your comments, we have rewritten in our revised manuscript (Tracked Version) (Line 325).

Line 286-288: Please explain the marking representing significance in Figure 1.B.

Response：Thanks for your comments, we have added the sentence of “Marking * represents significant difference, *, P < 0.05; **, P < 0.01; ***, P < 0.001” in revised legends of Figure 4.B (Line 948-949).

Figure 4.C and 4.D: Please clarify the meaning of the different coloured outlines of the circles.

Response：Thanks for your comments. The yellow-green outlines of the circles represent the ST is the founder ST of one clonal complex (CC), and the other STs (with purple outlines of the circles) of the CC are derived from the founder ST with two allelic differences. We have added the meaning of the different colored outlines of the circles in our revised Figure legends (Line 954-957).

Figure 4.E: Please clarify on which data the tree is based.

Response：Thanks for your comments. The Heatmap was performed by “pheatmap” in R-studio based on the distribution of phylogenetic groups and MLST for MDR strains in our study. The distribution of phylogenetic groups and MLST was transformed to “0” or “1”. The partial data of the transformation results are shown in the table below. For example, data for CD01 in the table below means the strain belonged to B2 group and ST10, CD02 in the table below means the strain belonged to D group and ST58.

NAME B1 B2 D ST10 ST58 ST361 ST1276

CD01 0 1 0 1 0 0 0

CD02 0 0 1 0 1 0 0

CD03 0 1 0 0 0 1 0

CD04 0 1 0 1 0 0 0

CD05 0 0 1 0 0 0 1

CD06 0 1 0 0 0 0 0

……

Line 368-369: You stated you found blaTEM-1, blaTEM-135, and blaTEM-176. Neither of these is a ESBL-encoding gene. Please rephrase the sentence accordingly.

Response：Thanks for your comments. We have checked the whole manuscript and changed the ESBL-encoding gene to β-lactam resistance genes in order to avoid confusion (Line 439).

Line379-380: Please add in Material and Methods, how you located the blaCTX-M-14 on the plasmid and present the data in results.

Response：Thanks for your comments. In our present study, we did not investigate the location of blaCTX-M-14. This sentence was proposed based on the references which provided to indicate the potential transmission risk of blaCTX-M-14 strains. And the references have been added at the end of the sentence (Line 449-450).

Line 383-388: The authors use very general terms in the two sentences. Please clarify, what “paying more attention” means. Please explain in more detail what you mean by “diversity and evolution” and the implications of that, or refrain from using the sentence.

Response：Thanks for your comments. In our present study, the high prevalence of blaCTX-M-14-positive and blaCTX-M-55-positive strains was observed, which are considered as high-risk strains for transmission between different hosts, thus, “paying more attention” means that more studies should focus on the capability of clonal transmission between different hosts for these strains in the future. We have added the corresponding content in our revised manuscript (Tracked Version) (Line 458-460).

Moreover, the other uncommon variants of blaCTX-M were not the focus of our discussion, so we have deleted the “diversity and evolution” in our revised manuscript.

Line 399-400: The authors are repeating the sentence from 391-392. Please amend.

Response：Thanks for your comments. We have deleted the repeating sentence in our revised manuscript.

Line 405-406: The authors speak in very general terms. The manuscript would improve by the use of examples.

Response：Thanks for your comments. We have modified the sentence in our revised manuscript (Tracked Version) (Line 480-485): The similar phenomenon was also found in E. coli from captive non-human primates and waterfowl in China which indicated the different expression of ARGs, such as the abnormal expression of ARGs and the expression of ARGs has not reached that level which can activate the antibiotic resistance, may be a possible reason the incomplete consistency between AMR and ARGs.

Line 412: Please cross-reference the percentages presented with the sources.

Response：Thanks for your comments. We have checked the percentages in reference and changed the sentence to “the ranging from 0.6% to 81.6% in Shaanxi and 2.53% to 87.34% in Shandong, respectively” in our revised manuscript (Tracked Version) (Line 501).

Line 412-414: As in the cited study a different set of virulence genes was tested, a comparison with the results of the study appears difficult. Please revise or delete.

Response：Thanks for your comments. We have deleted the sentence in our revised manuscript.

Line 416-419: Since the study does not provide data on the prevalence of combining VAGs separated into categories of ExPEC-related and DEC-related VAGs, it is difficult to follow the argument of the sentence.

Response：Thanks for your comments. In our present study, we separated the VAGs into categories of ExPEC-related VAGs (17–20) and DEC-related VAGs (21–23) according to previous studies. And 75.68% (56/74) of the strains carried at least one DEC-related VAG, which also carried at least one ExPEC-related VAG. We have added the corresponding data of strains combining ExPEC-related and DEC-related VAGs simultaneously in the “Results” section of our revised manuscript (Tracked Version) (Line 254-255).

Line: 428-429: The authors repeat the results already given in discussion. Please delete.

Response：Thanks for your comments. We have deleted the repeating sentence in our revised manuscript. 

Line 431-434: The authors explain the possible reason for the positive correlation between sul2 and ompT very well. Is there a similar rationale for the other two positive pairs? Is anything known about possible reasons for the negative association pairs?

Response 4：Thanks for your comments. We have added corresponding content in discussion for the possible reasons for the other two positive pairs and negative association pairs in our revise manuscript (Tracked Version) (Line 531-543): Similarly, the location of aacC, which usually was detected in the variable region of integron (24), may be related to the other two positive associations (aacC2/ompT, aacC4/astA) observed in our present study.

For the negative associations, the gene incompatibilities may be one of possible reasons according to previous researches (5,25). For example, the VAG cvaC usually be found in virulent IncF plasmids (9), while blaCTX-M is usually found in integron (10), the different location on mobile genetic elements may be related to the reason for negative association between blaCTX-M and cvaC observed in our present study. While the specific mechanism for the associations observed in our present study required more comprehensive studies to validate.

Line 442-443: Please clarify the sentence to avoid the reader’s confusion.

Response：Thanks for your comments. We have modified the sentence in our revised manuscript (Tracked Version) (Line 556-559): our present results showed a high prevalence of strains which belonged to virulent extraintestinal-related groups (B2 and D) were detected among MDR E. coli from pet dogs.

Line 448-449: Please clarify the sentence to avoid the reader’s confusion.

Response：Thanks for your comments. We have modified the sentence to “These phenomena indicated that the antimicrobial resistance may increase the possibility for carrying more virulent factor (for example, the number of VAGs carried) in E. coli strains” in our revised manuscript (Tracked Version) (Line 570-572). Actually, the similar phenomenon was found in previous study, which indicated that the probability of identifying virulence factor, such as STb, VT2, AIDA, was higher in strains carrying dhfrXIII, dhfrI, and tetB, respectively (5).

Line 452: According to the reference, ST131 has been identified the most common sequence type in ExPEC. Please rephrase.

Response：Thanks for your comments. We have modified the sentence to “which has been identified as the most common ST in ExPEC, covering all geographic regions” in our revised manuscript (Tracked Version) (Line 583).

Line 455: The reader might wonder as ST10 is the most prevalent sequence type in the study, if there is anything known about its distribution?

Response：Thanks for your comments. Actually, ST10 also was widely detected in E. coli from other animals in China, such as pigs (3,26) and chickens (1), indicating that the ST10 may be the prevalent ST in E. coli strains from animals in China. And we have also added the corresponding content in our revised manuscript (Tracked Version) (Line 591-593).

Line 462-463: Please specify.

Response：Thanks for your comments. We have changed the sentence to “Seventy-four MDR strains were observed among 135 E. coli isolates from pet dogs, of which 12 ARGs and 14 VAGs were detected, implying that the E. coli isolates can be considered as a pool of ARGs and VAGs” in our revised manuscript (Tracked Version) (Line 601-604).

 

Response to Reviewer 2 Comments

Response to Reviewer:

 Thank you very much for your time and thoughtful comments, many of which have been incorporated into the revised manuscript (Tracked Version). Detailed responses are as follows.

Reviewer #2: 

(1) Abstract: So far I know, any abstract has four essential parts with few additional or optional parts, such as objective(s), methodology, findings, and concluding remarks as essential parts and background, limitations, future indication etc could be present as additional portion(s). In this abstract, the objectives and methodology were not properly written, even absent. The abstract should be re-written properly, concisely and with proper structures.

Response 1：Thanks for your comments. As the reviewer suggested, we have modified the “Abstract” in our revised manuscript (Tracked Version) (Line 15-41).

(2) Introduction: In my opinion, research gaps, justification and novelty of the present research were not mentioned or presented properly.

Response 2：Thanks for your comments. Recently, increasing studies have focused on antimicrobial resistant E. coli from companion animals, indicating that the companion animals are reservoirs of MDR strains with a high risk of transmission between dogs and humans (27). However, the research on MDR strains from companion animals in China is limited, especially in Sichuan Province, which is considered to be a major province of pets keeping in western China (28). In our present study, we not only analyze the antibiotic resistance of E. coli isolated from pet dogs, but also pay more attention to the ARGs, VAGs and population structure (ST and phylogenetic groups) for MDR strains to evaluate the potential threat. Moreover, more and more studies have observed the association which existed in between ARGs and VAGs in E. coli from animals, such as pandas (29) and waterfowl (19), but studies for the association between ARGs and VAGs of E. coli from pet dogs was limited, we further analyzed the association between ARGs and VAGs in MDR strains. Above all, our research helps to better understand the characteristics of MDR strains from pet dogs and fills the research gaps of MDR E. coli from pet dogs in Sichuan Province. And we have modified the “Introduction” section in our revised manuscript (Tracked Version) (Line 46-90).

(3) Methodology:

(A) Sample size could be larger, and could be determined using formula. However, the less number of samples could be explained properly in the 2.1 section.

Response 3A: Thanks for your comments. Actually, we collected about 250 fecal samples in the initial phase of the present study, while some of the samples did not meet the sampling standards for “Samples were excluded if the pet dogs were prescribed antimicrobial therapy or veterinary admission within the previous 3 months” in our present study. The sampling standards were also shown in 2.1 section (Line 83-84). Thus, only 185 fecal samples were ultimately included. We have added the explanation for limited number of samples in the “Limitations” section in our revised manuscript (Tracked Version) (Line 612-616).

(B) Line 77: why this findings unnecessarily came here?

Response 3B: Thanks for your comments. We have deleted this finding in 2.1 section and added it to “Results” section in our revised manuscript (Tracked Version) (Line 199-200).

Line 80-81: Could you please provide the permission number?

Response: Thanks for your comments. The permission number is DYY-2020303164, and we have added it in our revised manuscript (Tracked Version) (Line 100).

(C) strain isolation portion (section 2.2) was made unnecessarily complex.

Response 3C: Thanks for your comments. We have simplified the section 2.2 in our revised manuscript (Tracked Version) (Line 111-127): The isolation and identification of E. coli was performed as previous studies described. Fecal samples were enriched in Luria-Bertani (LB) broth at 37 °C, 120 r/min in a shaking incubator for 24 h. All isolates were identified presumptively by using phenotypic methods, including Gram staining, MacConkey agar growth and Eosin Methylene Blue agar growth. We further used 16S rDNA sequences (Primer: 5’-GAGTTTGATCCTGGCTCAG-3’; 5’-AGAAAGGAGGTGATCCAGCC-3’) for final identification of E. coli. The confirmed isolates were stored in Luria-Bertani (LB) broth containing 50% glycerol at −20 °C for further analysis.

(D) Line 113-115: these information could be added in discussion.

Response 3D: Thanks for your comments. Actually, the information is the criteria for our selection of antimicrobial agents for susceptibility testing in our present study, it is more scientific to place the information in “2.3” section, and we have modified the sentence to“For the 16 antimicrobial agents, the CN and TOB in aminoglycosides, DOX in tetracyclines, KZ, CTX, AMP, AMC in β-lactams and CIP in quinolones were used in pet animals at the location of our present study. The other antimicrobial agents (TET, C, CXM, FEP, ATM, IPM, FOX and SXT) have been found with E. coli resistance in dogs according to previous studies” in our revised manuscript (Tracked Version) (Line 139-143).

(E) Line 124-125: please rephrase the sentence.

Response 3E: Thanks for your comments. We have rephrased the sentence to “Primers of 23 ARGs (including 5 blaCTX-M alleles for group1, 2, 8, 9 and 25) in 6 categories and 25 VAGs in 5 categories were synthesized by Huada Gene Technology Co. Ltd (Shenzhen, China), and the PCR primer and conditions for the ARGs and VAGs which been chosen in our present study were showed in S1 Table and S2 Table, respectively” in our revised ma

---

## [Editor Report · Decision Letter 1]

17 Jan 2024

Characteristics of MDR E. coli strains isolated from Pet Dogs with clinic diarrhea: A Pool of Antibiotic Resistance Genes and Virulence-Associated Genes

PONE-D-23-30998R1

Dear Dr. Zhong,

We’re pleased to inform you that your manuscript has been judged scientifically suitable for publication and will be formally accepted for publication once it meets all outstanding technical requirements.

Kind regards,

Professor Md. Tanvir Rahman, DVM, MSc, PhD

Academic Editor

PLOS ONE

Additional Editor Comments (optional):

Thanks for addressing all the comments of the reviewers.
---

## [Editor Report · Acceptance letter]

5 Feb 2024

PONE-D-23-30998R1 

PLOS ONE

Dear Dr. Zhong, 

I'm pleased to inform you that your manuscript has been deemed suitable for publication in PLOS ONE. Congratulations! Your manuscript is now being handed over to our production team.

Kind regards, 

on behalf of

Professor Md. Tanvir Rahman 

Academic Editor

PLOS ONE